

# Dietary reference intake for military operations: a scoping review

Ryoko Mizushima[1,2], Motohiko Miyachi[3,4], Eiichi Yoshimura[5], Yoichi Hatamoto[5], Mai Matsumoto[2], Yuka Hamada[5], Mana Hatanaka[5], Aya Maeno[5], Chifumi Shimomura[2] and Hidemi Takimoto[2]

[1] Faculty of Health and Sport Sciences, University of Tsukuba, Tsukuba, Ibaraki, Japan
[2] Department of Nutritional Epidemiology and Shokuiku, National Institutes of Biomedical Innovation, Health, and Nutrition, Settsu, Osaka, Japan
[3] Faculty of Sport Sciences, Waseda University, Tokorozawa, Saitama, Japan
[4] Department of Physical Activity Research, National Institutes of Biomedical Innovation, Health, and Nutrition, Settsu, Osaka, Japan
[5] Department of Nutrition and Metabolism, National Institutes of Biomedical Innovation, Health and Nutrition, Settsu, Osaka, Japan

Corresponding author
Motohiko Miyachi,
miyachim@waseda.jp

## ABSTRACT

**Background**. Reports that collect and organize dietary reference intake (DRI) data for military operations in different countries and regions worldwide are limited. This scoping review aimed to collect and organize information on the status of formulating a DRI for military operations in each country.

**Methodology**. For the information search, we queried PubMed and Google for literature and reports on the DRI for military operations and summarized the content of the adopted literature and reports.

**Results**. The content and rationale for DRI for military operations in Australia, the United Kingdom (UK), the United States of America (USA), and the North Atlantic Treaty Organization (NATO) can be summarized as follows: (1) Energy requirements: Four reports formulated physical activity levels (PALs) and corresponding energy requirements that differed from those for the civilian public. The PAL range for the military was set as high as 1.50–3.20, as opposed to the standard civilian upper PAL set at 1.20–2.20. (2) Protein: Three military reports outside of the UK had different standards than those for the civilian public with an increased intake in accordance with the high PAL while simultaneously preventing excessive intake. In the military, values were formulated 1.2–4.8 times higher than the standards for civilians (45–65 g/day to 55–307 g/day). (3) Macronutrient energy distribution: Four military reports established macronutrient energy distributions that differed from those for the civilian public. The DRI for the Australian and UK militaries was formulated such that as PAL increased, protein decreased, fat decreased or remained unchanged, and carbohydrate increased. (4) Sodium: Considering that military personnel sweat more due to high physical activity and their environment, two Australian and NATO reports were established with sodium levels that were twice as high as that of the civilian public (460–2,300 mg/day to 920–3,200 mg/day). Increasing sodium intake to <4,800 mg/day is recommended for individuals who sweat a lot or are not accustomed to hot environments.

**Conclusions**. The DRI in Australia, the UK, USA, and NATO consider the physical activity and operating environment of military personnel, differing from those of the

civilian population in terms of (1) energy requirements, (2) protein, (3) macronutrient energy distribution, and (4) sodium.

# INTRODUCTION

Dietary reference intakes (DRIs) indicate standards for energy and nutrient intake to maintain and promote health and prevent lifestyle-related diseases. DRIs are primarily aimed at healthy individuals and groups (*Department of Health, 1991*; *German Society for Nutrition e. v, 2019*; *IOM, 1997*; *NHMRC, 2017*; *Nordic, 2014*; *Scientific Advisory Committee on Nutrition, 2011*; *Scientific Advisory Committee on Nutrition, 2015*; *Scientific Advisory Committee on Nutrition, 2016a*; *Subcommittee on the Tenth Edition of the RDAs, 1989*; *WHO, 2005*). Appropriate standards for energy and nutrient requirements for specific individuals and populations have been developed in various countries and are used in dietary guidelines for meal planning and evaluation (*Barr et al., 2003*; *IOM, 1997*). These standards are based on scientific evidence, considering factors such as age, sex, amount of activity, presence or absence of pregnancy or menstruation, and disease status. These standards were primarily designed for civilians without considering special groups, such as athletes and individuals with occupations requiring high levels of physical activity.

Energy and nutrient intake commensurate with the amount of training is extremely important for athletes and those in physically intense occupations (*Ahmed et al., 2023*; *Garron & Klein, 2023*; *Heaney et al., 2010*; *Johnson & Mayer, 2020*; *Vermeulen, Boyd & Spriet, 2021*). In particular, military personnel are required to be physically and mentally strong (*Ahmed et al., 2023*; *Garron & Klein, 2023*), requiring them to consume a diet that supports their duties (*Barringer et al., 2018*; *Chapman et al., 2021*; *Collins et al., 2020*; *Vyas & Cialdella-Kam, 2020*). Inadequate nutritional intake during training negatively impacts military personnel's health, physical fitness, and performance (*Chapman et al., 2020*; *Murphy et al., 2018*; *O'Leary, Wardle & Greeves, 2020*). Therefore, previous research has examined energy and nutrient requirements appropriate for the high physical activity and extreme environments associated with military activities (*Baker et al., 2020*; *Chapman et al., 2019*; *Chapman et al., 2020*; *Chapman et al., 2021*; *Collins et al., 2020*; *Herzman-Harari et al., 2013*; *Lutz et al., 2019*; *O'Leary, Wardle & Greeves, 2020*; *Vyas & Cialdella-Kam, 2020*).

The scoping review by *Collins et al. (2020)* organized methods for surveying dietary intake among military personnel and those retired and reported whether comparisons of dietary intake obtained from dietary surveys with dietary guidelines were included in each study. The 89 articles in their review assessed dietary intake in military settings, primarily using food frequency questionnaires (FFQs) or 24-hour recall. In addition, 54 (61%) articles compared their diets to the national dietary guidelines. Of these, 39 (44%) were compared with dietary guidelines for the civilian population, one (1%) with sports

nutrition guidelines, and 14 (16%) with military guidelines. This included the DRI for military operations utilized (*Herzman-Harari et al., 2013*).

The USA's DRI is typically applied in these studies for military operations (*Baker-Fulco et al., 2001*; *Department of Defense, 2001*; *Department of Defense, 2017*; *IOM, 1994*). Specifically, the Military Recommended Dietary Allowances (MRDA) (*IOM, 1994*) was established in 1985, and the Military Dietary Reference Intakes (MDRIs) (*Baker-Fulco et al., 2001*; *Department of Defense, 2001*) was revised in 2001. A recent study reported using the 2017 version of the MDRI (*Department of Defense, 2017*) to assess the diet of military personnel (*Lutz et al., 2019*). Studies using the Military Dietary Reference Value (Military DRV) (*Casey, 2008*; *Scientific Advisory Committee on Nutrition, 2016b*; *SACN, 2020*) in the United Kingdom (UK) have also been reported (*Chapman et al., 2019*).

To our knowledge, the DRI for military operations (*Baker-Fulco et al., 2001*; *Casey, 2008*; *Department of Defense, 2001*; *Department of Defense, 2017*; *Scientific Advisory Committee on Nutrition, 2016b*; *SACN, 2020*; *IOM, 1994*) are among the few guidelines that are appropriate for a special population with high physical activity and physical and mental strength requirements. Therefore, it is also used to compare the dietary intake status of athletes (*Heaney et al., 2010*) and firefighters (*Johnson & Mayer, 2020*) who perform high physical activity. However, only a limited number of reports have collected and organized the DRI for military operations in different countries and regions worldwide.

Therefore, the current study seeks to compare the DRIs for military operations in several countries and regions with those for the civilian population through a scoping review.

## SURVEY METHODOLOGY

A scoping review, which does not have a clear hypothesis, is appropriate for this topic (*Peters et al., 2020*). The review procedure involved three stages: (1) a literature survey on DRI for military operations using PubMed; (2) screening DRI information for military operations (survey reports, materials, *etc.*) using Google browser; and (3) organizing the contents of survey reports and documents describing DRI for military operations.

### Search for literature on DRI for military operations using PubMed

A search was conducted using PubMed on May 31, 2020, using the following search formula (military[TIAB] OR soldier*[TIAB] OR force*[TIAB] OR levy*[TIAB] OR levie*[TIAB] OR ''active duty''[TIAB] OR ''active duties''[TIAB] OR ''armed service*''[TIAB] OR ''armed force*''[TIAB] OR ''defense service*''[TIAB] OR ''defense force*''[TIAB] OR army[TIAB] OR armies) AND ((diet* [TIAB] OR food* [TIAB] OR consumption* [TIAB] OR intake* [TIAB] OR nutrition* [TIAB] OR nutrient* [TIAB] OR energy [TIAB])) AND (AND ''scoping review'' AND (''1879/01/01''[PDAT]: ''2020/05/31''[PDAT])). The inclusion criteria for the academic studies were: (1) inclusion of military personnel, (2) description of the characteristics of the target population, (3) written in English, and (4) research on energy and nutrient requirements and dietary intake. This search aimed to extract the literature on DRI for military operations.

## Search for information on DRI for military operations (survey reports, materials, *etc*.) using the Google browser

Academic literature regarding DRI for military operations in each country is lacking, and the methodology in Step 1 alone is insufficient. Therefore, to obtain other materials, such as investigative reports, we conducted an Internet search using the Google browser. The DRI for military operations in each country was searched using keywords such as the name of the target country, "military", and "dietary reference intake".

The target countries were the 16 countries identified in the results of the search in Step 1 that were covered in the review by *Collins et al. (2020)* as well as the Netherlands, North Atlantic Treaty Organization (NATO), Denmark, and Ireland, which were mentioned in the text, and Sweden, totaling 20 countries and one institution, of which, only four were from Asia. Thus, we opted to include a report targeting Asians in addition to the 20 countries and one institution due to the marked differences between Westerner and Asian physiques. Therefore, with reference to the World Bank Country and Lending Groups Country Classification (*The World Bank, 2019*), we investigated the dietary intake standards for the militaries of 37 countries and one organization, including 17 countries and regions in East and Southeast Asia. The last search date was September 30, 2020. Subsequently, we narrowed down the survey reports published by planning organizations.

## Organizing the contents of survey reports and documents that describe DRI for military operations

We read the descriptions in the final selected survey reports and excluded those that lacked sufficient information to organize the details of the DRI formulations and rationale. Specifically, we adopted documents that described standards for energy and nutrients and the rationale for their formulation, as well as DRI for the civilian population, and excluded documents that only described standards for energy and macronutrient energy distribution. Two authors (RM and HT) decided to accept or reject the article after discussion. The DRI for civilians adopted the reports cited in each DRI for military operations.

## RESULTS AND DISCUSSION

### Search results

#### *Search for literature on DRI for military operations using PubMed*

The search yielded eight references, seven of which were excluded as they did not target the military. The accepted reference (*Collins et al., 2020*) was a scoping review of dietary assessment methods in military personnel and veterans. A keen study of this scoping review by *Collins et al. (2020)* revealed survey reports of dietary intake standards for the military in the USA (*Baker-Fulco et al., 2001*; *Department of Defense, 2001*; *Department of Defense, 2017*) and Australia (*Forbes-Ewan & DSTO, 2009*). Moreover, the Nordic countries (Denmark, Finland, Ireland, Norway, and Sweden) utilize the DRI (*NATO, 2010*) developed by NATO. Meanwhile, another report (*Valk & Pasman, 2005*) described dietary intake in the Dutch military.

*Collins et al.'s (2020)* scoping review selected 89 research reports from 16 countries (USA, Belgium, Finland, UK, Norway, Italy, Greece, France, Israel, Iran, Malaysia, Japan,

Australia, Cameroon, South Africa, and Brazil). In addition, since only research reports written in English were accepted, only four countries in Asia (Israel, Iran, Malaysia, and Japan) were represented.

### Search for information on DRI for military operations (survey reports, materials, etc.) using the Google browser

A Google search was conducted targeting 37 countries and one institution, including the countries identified in *Collins et al. (2020)* review, as well as countries in East Asia and Southeast Asia. As a result, survey reports and related materials on eight DRI for military operations for eight militaries (USA, Australia, NATO, Netherlands, UK, China, South Korea, and Singapore) were found (*Chan, 2016*; *Department of Defense, 2017*; *Forbes-Ewan & DSTO, 2009*; *Guo et al., 2016*; *MND, 2017*; *NATO, 2010*; *Scientific Advisory Committee on Nutrition, 2016b*; *Valk & Pasman, 2005*).

### Organizing the contents of research reports and documents that describe DRI for military operations

After carefully reading the contents of the research reports that were finally selected, the Dutch (*Valk & Pasman, 2005*) survey report had "Content describing protocol limitations and protocol details for testing the effects of dietary interventions in military settings". Three countries, Singapore (*Chan, 2016*), China (*Guo et al., 2016*), and South Korea (*MND, 2017*), were not included in the review due to insufficient description of the basis for their formulation and were instead used as reference materials. Finally, we scrutinized the DRI for the military operations of three countries and one organization: Australia, UK, USA, and NATO (*Department of Defense, 2017*; *Forbes-Ewan & DSTO, 2009*; *NATO, 2010*; *Scientific Advisory Committee on Nutrition, 2016b*). We referred to the unrevised DRI (*Baker-Fulco et al., 2001*; *Casey, 2008*; *Department of Defense, 2001*; *Forbes-Ewan, Materials Research Laboratory & Defence Science and Technology Organisation (DSTO), 1993*; *Forbes-Ewan, DSTO-Scottsdale & CBRN Defence Centre Platform Sciences Laboratory, 2002*; *SACN, 2020*; *IOM, 1994*), the DRI for the civilian public (*Department of Health, 1991*; *IOM, 1997*; *NHMRC, 2017*; *Scientific Advisory Committee on Nutrition, 2011*; *Scientific Advisory Committee on Nutrition, 2015*; *Scientific Advisory Committee on Nutrition, 2016a*; *Subcommittee on the Tenth Edition of the RDAs, 1989*), and materials citing these. Based on this, we summarized four published DRIs for military operations, outlined the basics of their formulation, and organized the standards for energy and nutrients that had been formulated.

## Overview of four dietary reference intakes for military operations

The contents of the survey reports for three countries and one organization, and the corresponding survey reports on DRI for the civilian public (*Department of Health, 1991*; *IOM, 1997*; *NHMRC, 2017*; *Scientific Advisory Committee on Nutrition, 2011*; *Scientific Advisory Committee on Nutrition, 2015*; *Scientific Advisory Committee on Nutrition, 2016a*; *Subcommittee on the Tenth Edition of the RDAs, 1989*) were organized (Table 1). Three countries, namely, Australia, the UK, and the USA (*Department of Defense, 2017*; *Forbes-Ewan & DSTO, 2009*; *Scientific Advisory Committee on Nutrition, 2016b*), have had at least

**Table 1  Overview of the development and update timeline for civilian and military dietary reference intakes (DRI).**

| | | Australia | UK | USA | NATO |
|---|---|---|---|---|---|
| **Civilian**[a] | Year formulated | 2006, 2017 | 1991, 2011, 2015, 2016 | 1989, 1997, 1998, 2000, 2001, 2005, 2011, 2019 | 2006, 2017 |
| | Formulating organization | National Health and Medical Research Council; NHMRC | (1) British Nutrition Foundation and Committee on Medical Aspects of Food and Nutrition Policy (Until 1991) (2) British Nutrition Foundation and Scientific Advisory Committee on Nutrition; SACN | (1) Subcommittee on the Tenth Edition of the RDAs (2) Food and Nutrition Board of the Institute of Medicine; FNB | National Health and Medical Research Council; NHMRC |
| | Name of report or book | Nutrient Reference Values for Australia and New Zealand, Including Recommended Dietary Intakes | Dietary Reference Values for Food Energy and Nutrients for the United Kingdom (1991), Dietary Reference Values for Energy (2011), Carbohydrates and Health (2015), Vitamin D and Health (2016) | Recommended Dietary Allowances, Dietary Reference Intakes; DRIs: Calcium, Phosphorus, Magnesium, Vitamin D, and Fluoride (1997), Thiamin, Riboflavin, Niacin, Vitamin B6, Folate, Vitamin B12, Pantothenic Acid, Biotin, and Choline (1998), Vitamin C, Vitamin E, Selenium, and Carotenoids (2000), Vitamin A, Vitamin K, Arsenic, Boron, Chromium, Copper, Iodine, Iron, Manganese, Molybdenum, Nickel, Silicon, Vanadium, and Zinc (2001), Energy, Carbohydrate, Fiber, Fat, Fatty Acids, Cholesterol, Protein, and Amino Acids (2005), Water, Potassium, Sodium, Chloride, and Sulfate (2005), Calcium and Vitamin D (2011), Sodium and Potassium (2019) | Nutrient Reference Values for Australia and New Zealand, Including Recommended Dietary Intakes[c] |
| | Language | | | English | |
| **Military** | Year formulated | 1993, 2002, 2009 | 2008, 2016, 2020 | 1985, 2001, 2017 | 2010 |
| | Formulating organization | Defence Science and Technology Organisation; DSTO | (1) QinetiQ Ltd. (2) Scientific Advisory Committee on Nutrition; SACN | Headquarters Departments of the Army, the Navy, and the Air Force | North Atlantic Treaty Organization; NATO (Wageningen University and Research Centre, Netherlands) |
| | Name of report[b] | Australian Defence Force Nutritional Requirements in the 21st Century (Version 1) | (1) Military Dietary Reference Values. Farnborough (2) Statement on Military Dietary Reference Values for Energy | Nutrition and Menu Standards for Human Performamce Optimization | Nutrition Science and Food Standards for Military Operations (Recommendations for nutrient composition of combat rations for the NATO response force) |
| | Language | | | English | |

**Notes.**

[a]The civilian DRIs were used as the precursor for DRI development for military operations.

[b]Nutrients are published as they are formulated.

[c]NATO referred primarily to "Nutrient Reference Values for Australia and New Zealand, Including Recommended Dietary Intakes" as the DRI for civilians.

one revision in the past (*Baker-Fulco et al., 2001*; *Casey, 2008*; *Department of Defense, 2001*; *Forbes-Ewan, Materials Research Laboratory & Defence Science and Technology Organisation (DSTO), 1993*; *Forbes-Ewan, DSTO-Scottsdale & CBRN Defence Centre Platform Sciences Laboratory, 2002*; *SACN, 2020*). In developing NATO's DRI (Nutrient Intake Value; NIV), reference was made to the DRI for civilians in Australia, New Zealand (*NHMRC, 2017*), Germany, Austria, Switzerland (DACH, 2019), the USA, Canada (*IOM, 1997*), the Nordic countries (*Nordic, 2014*), and WHO *WHO, 2005*). Among these, the National Health and Medical Research Council's (*NHMRC, 2017*) Nutrient Reference Values (NRVs), the DRI for Australia and New Zealand, were considered the main reference material as they are the most recent and are formulated based on scientific evidence (Tables 1 and 2).

**Table 2** Overview of dietary reference intakes (DRI) indicators in the United States of America (USA) and Canada and corresponding DRI indicators for civilians.

| Name | USA | | Australia NRV | UK DRV | NATO NIVs |
|---|---|---|---|---|---|
| | Dietary reference intake (DRI) Indicator Acronym and Description | | | | |
| | Energy | | | | |
| | EER | | EER[a] | EAR[a] | ANR |
| | Nutrients | | | | |
| | EAR | An average daily nutrient intake level estimated to meet the requirement of half the healthy individuals in a particular life stage and gender group. | EAR | EAR | ANR |
| | RDA | An average daily level of intake sufficient to meet the nutrient requirements of nearly all (97%–98%) healthy people. | RDI | RNI | INL$_{97.5}$[b] |
| | AI | An established when evidence is insufficient to develop an RDA and is set at a level assumed to ensure nutritional adequacy. | AI | Safe intakes | AI |
| Applicable in USA | UL | A maximum daily intake unlikely to cause adverse health effects. | UL | — | — |
| | AMDR | An estimate of the range of intake for each macronutrient for individuals (expressed as per cent contribution to energy) | AMDR | Population average | AMDR |
| | CDRR | The lowest level of intake for which there was sufficient strength of evidence to characterize a chronic disease risk reduction. | — | — | — |
| Not applicable in USA | Others[+] | | SDT[+] | LRNI[++] , DRV[+++] | — |

**Notes.**

[a] EER in Australia and USA: Values for estimating maintenance of energy balance by age, gender, body size, and PAL level

[b] INL$_{97.5}$: Specifically, it was defined as 97.5% instead of 97–98%

BMI = body weight(kg) ÷ (body height (m))$^2$

Other indicators not applicable in the USA:

[+] SDT: Daily intake from certain foods and beverages that may help prevent chronic diseases.

[++] LRNI: It is the lower limit nutrient intake standard and is one of the indicators in the UK's DRV.

[+++] DRV: Regarding dietary fiber in the UK's DRV, it is considered that it does not correspond to any indicator due to lack of evidence, so DRV is used.

Abbreviation: AI, Adequate Intake; AMDR, Acceptable Macronutrient Distribution Range; ANR, Average Nutrient Requirement; BMI, Body Mass Index; CDRR, Chronic Disease Risk Reduction Intake; DRI, Dietary Reference Intakes; DRV, Dietary Reference Value; EAR, Estimated Average Requirement; EER, Estimated Energy Requirement; INL97.5, Individual Nutrient Level 97.5; LRNI, Lower Reference Nutrient Intake; NRV, Nutrient Reference Values; RDA, Recommended Dietary Allowance; RDI, Recommended Dietary Intake; RNI, Reference Nutrient Intake; SDT, Suggested Dietary Target; UL, Tolerable Upper Intake Level.

## Overview of four published DRIs for military operations
### Established energy and nutrients in the four published DRIs for military operations

All four published DRIs for military operation formulations were based on the referenced DRIs for civilians; however, standards were set by adding the amount necessary for members who have a high level of physical activity and are responsible for military activities in special environments (*Department of Defense, 2017*; *Forbes-Ewan & DSTO, 2009*; *NATO, 2010*; *Scientific Advisory Committee on Nutrition, 2016b*). The basis for determining the required intake in high-intensity physical activity and special environments was (1) research reports, survey reports, *etc.*, describing research findings (*Askew, 1995*; *IOM, 1996*; *Rodriguez, Di Marco & Langley, 2009*; *Tarnopolsky et al., 2005*; *Tomczak et al., 2016*) and (2) data on the physical characteristics of the military members, their basal metabolic rates,

**Table 3  Dietary reference intakes (DRI) for military operations comparison to civilian norms.**

| | | Australia | UK | USA | NATO |
|---|---|:---:|:---:|:---:|:---:|
| | Energy (kcal) | ◎ | ◎ | ◎ | ◎ |
| | Macronutrient energy distribution (%) | ◎ | ◎ | ◎ | ◎ |
| | Protein (g) | ◎ | | ◎ | ◎ |
| | Fat (g) | | | ◎ | ◎ |
| Carbohydrate | Carbohydrate (g) | ◎ | | ◎ | ◎ |
| | Dietary fiber (g) | ○ | △ | ◎ | ○ |
| Vitamins | Fat-soluble — Vitamin A (μg) | ○ | △ | ○ | ○ |
| | Vitamin D (μg) | ○ | △ | ○ | ○ |
| | Vitamin E (mg) | ○ | △ | ○ | ○ |
| | Vitamin K (μg) | ○ | △ | ○ | ○ |
| | Water-soluble — Vitamin B$_1$ (mg) | ◎ | △ | ○ | ○ |
| | Vitamin B$_2$ (mg) | ◎ | △ | ○ | ◎ |
| | Niacin (mg) | ◎ | △ | ○ | △ |
| | Vitamin B$_6$ (mg) | ◎ | △ | ○ | ◎ |
| | Vitamin B$_{12}$ (μg) | ○ | △ | ○ | ○ |
| | Folic acid (μg) | ○ | △ | ○ | ○ |
| | Pantothenic acid (mg) | ○ | △ | △ | ○ |
| | Biotin (μg) | ○ | △ | △ | ○ |
| | Vitamin C (mg) | ○ | △ | ○ | ○ |
| Minerals | Macro — Sodium (mg) | ◎ | △ | ◎ | ◎ |
| | Potassium (mg) | ○ | △ | ○ | ○ |
| | Calcium (mg) | ○ | △ | ○ | ○ |
| | Magnesium (mg) | ○ | △ | ○[*] | ○[*] |
| | Phosphorus (mg) | ○ | △ | ○ | ○ |
| | Micro — Iron (mg) | ○ | △ | ○ | ◎ |
| | Zinc (mg) | ○ | △ | ○ | ◎ |
| | Copper (mg) | ○ | △ | △ | ◎ |
| | Manganese (mg) | ○ | △ | △ | ○ |
| | Iodine (μg) | ○ | △ | ○ | ○ |
| | Selenium (μg) | ○ | △ | ○ | ○ |
| | Chromium (μg) | ○ | △ | △ | ○ |
| | Molybdenum (μg) | ○ | △ | △ | ○ |

**Notes.**

The nutrient items described in this review were assumed to be common to the four published DRI reports for Military Operations

◎: The DRI for Military Operations has been formulated based on scientific evidence, and is different from the DRI for civilians.

○: The DRI for Military Operations is based on scientific evidence and has been formulated to the same value as the DRI for civilians, and the DRI value for civilians is stated in the DRI for Military Operations survey report

△: The DRI for Military Operations has been formulated to be the same value as for civilians based on scientific evidence, but the DRI value for civilians is not listed in the DRI for Military Operations survey report

*The DRI for civilians was reflected, taking into consideration the target age category.

physical activity, *etc.* (*Baker-Fulco et al., 2001*; *Department of Defense, 2001*; *Forbes-Ewan et al., 1995*; *Paquette, Gordon & Bradtmiller, 2009*). Table 3 presents the comparison of DRIs for military operations.

Energy and nutrient indicators are important (*King, Vorster & Tome, 2007*) and are formulated and displayed in various ways (Tables 2 and S1). Therefore, we organized

**Table 4  Physical activity level (PAL) category in DRI for civilians and military.**

| | Civilian (PAL) | Military (PAL) |
|---|---|---|
| Australia | I: Bed rest (1.20)<br>II: Very sedentary (1.40)<br>III: Light activity (1.60)<br>IV: Moderate activity (1.80)<br>V: Heavy activity (2.00)<br>VI: Vigorous activity (2.20) | I: Low physical activity (1.50)<br>II: Moderately active (1.80)<br>III: Very active (2.10)<br>IV: Extreme activity (2.40)<br>V: Believed to approach the limit of<br>human endurance (3.20) |
| UK[a] | I: 25th percentile (Less active [1.49])<br>II: Median (Population [1.63])<br>III: 75th percentile (More active [1.78]) | I: General population (1.63)<br>II: Active service (2.08)<br>III: Military training courses A (2.32)<br>IV: Military training courses B (2.62) |
| USA[b] | I: Sedentary (1.25)<br>II: Low active (1.50)<br>III: Active (1.75)<br>IV: Very active (2.20) | I: General / routine<br>II: Light activity (1.70)<br>III: Moderate activity (1.80)<br>IV: Heavy activity (2.20)<br>V: Exceptionally-heavy activity (2.50) |
| NATO[c] | — | I: Civilian norm (1.60)<br>II: Normal operations (2.00)<br>III: Combat or special forces operations (2.40)<br>(Hot environment, Cold environment,<br>High–altitude environment) |

**Notes.**

[a]PAL values for under 19 years of age in the UK's civilian DRI were slightly different: 25th percentile (1.68), median (1.75) and 75th percentile (1.86).

Additionally, the DRI for Military Operations had a 25th percentile, median, and 75th percentile for each PAL category. III: Military training courses A is included the following training groups: the common military syllabus for recruits (CMS(R)); Royal Air Force (RAF) phase-1 recruits. IV: Military training courses B is included the following training groups: Common Infantry Course (CIC)–paras and Guards; Commissioning Course for Officer Cadets (CCOC); Section Commander's Battle Course (SCBC) (army infantry soldiers phase-3 training).

[b]I: Since there is no description of the PAL value for General/routine, the exact value is unknown, but the standard PAL for civilians is Secondary (1.25), Low Active (1.50), Active (1.75) and Very Active (2.20), a range of 1.25–1.50 is possible

[c]NATO's PAL Category was developed for Australia and New Zealand, Germany and Austria and Switzerland (2002), USA and Canada (1997–2005), and the Nordic countries (2004) and WHO (2004) DRI were used as reference.

Abbreviation: PAL, physical activity level.

the data based on the type of indicator (Table 3). Australia, the UK, and the USA have set new standards for their military based on indicators of DRI for civilians (*Department of Defense, 2017*; *Forbes-Ewan & DSTO, 2009*; *Scientific Advisory Committee on Nutrition, 2016b*). Moreover, NATO used the NIV by *King, Vorster & Tome (2007)*, which differs from the NRV (*NHMRC, 2017*) index (*NATO, 2010*).

PAL category definitions and estimated energy requirement calculations vary among nations/organizations by age, sex, reference body size, and activity level (Tables S2 and S3). Specific energy and nutrient DRIs for civilian and military populations by nation/organization are depicted in Supplementary Tables S4–S18. These supplementary tables were used to develop the consolidated framework needed to illustrate differences in DRIs between civilian and military populations, as provided in Table 3 and PAL categories in Table 4.

### PAL categories and energy requirements

Energy requirements were common to all four published DRIs for military operations (*Department of Defense, 2017*; *Forbes-Ewan & DSTO, 2009*; *NATO, 2010*; *Scientific Advisory*

*Committee on Nutrition, 2016b*), with PALs and corresponding energy requirements that reflected the high physical activity of military personnel and differed from those in the DRI for civilians (*Department of Health, 1991*; *IOM, 1997*; *NHMRC, 2017*; *Scientific Advisory Committee on Nutrition, 2011*; *Scientific Advisory Committee on Nutrition, 2015*; *Scientific Advisory Committee on Nutrition, 2016a*; *Subcommittee on the Tenth Edition of the RDAs, 1989*) (Table 3). The PAL categories and energy requirements in DRIs for civilians and military are presented in Table S2, while the reference body size and estimated formula in DRIs for military operations is shown in Table S3.

The range of PAL categories for civilians is 1.20–2.20 (*Department of Health, 1991*; *IOM, 1997*; *NHMRC, 2017*; *Scientific Advisory Committee on Nutrition, 2011*), whereas the DRI for military operations ranges from similar PAL categories (low physical activity (1.50), general population (1.63), *etc.*) to categories reflecting higher levels of physical activity associated with engaging in military activities (1.50–3.20 in the case of Australia) (*Department of Defense, 2017*; *Forbes-Ewan & DSTO, 2009*; *NATO, 2010*; *Scientific Advisory Committee on Nutrition, 2016b*). A meta-analysis by *Murphy et al. (2018)* showed that insufficient energy intake in military personnel reduces lower extremity performance during military operations. Moreover, energy deficiency among military personnel is associated with heart disease risk (*Hilgenberg et al., 2016*). In contrast, overweight and obesity among military personnel is also a concern (*Herzman-Harari et al., 2013*). This suggests that adequate intake of the required amount of energy is very important for military personnel. Therefore, for military personnel, proper intake of the necessary amount of energy is important.

Moreover, the range of energy consumption varies depending on the job type of military personnel (*Forbes-Ewan et al., 1990*; *Forbes-Ewan & Waters, 1991*; *Forbes-Ewan & DTSO, 2009*; *Tharion et al., 2005*), and the range is wider than that of civilians. *Tharion et al. (2005)* found that the energy expenditures measured in 424 military members across all four military services, in a variety of occupations, climates, and environments (field *vs.* garrison) ranged from 13.0–29.8 MJ (3,109–7,131 kcal)/day for male military members and 9.8–23.4 MJ (2,332–5,597 kcal)/day for female military members. The category of "Believed to approach the limit of human endurance" (PAL 3.20) in Australia was based on the energy consumption of 28 MJ (6,692 kcal)/day for members selected for Special Air Service Regiments. Furthermore, since these military members are actively moving their bodies for about 20 h a day, they have a special system in which they only get about four hours of sleep (*Erdman et al., 2006*).

In the DRI for military operations, not only the amount of physical activity but also the environment was considered, and target categories were established, and energy and nutrients were formulated accordingly (*Department of Defense, 2017*; *Forbes-Ewan & DSTO, 2009*; *NATO, 2010*; *Scientific Advisory Committee on Nutrition, 2016b*). In NATO's DRI (*NATO, 2010*), the environmental cut-off values in the review by *Askew (1995)* (hot environment: >30 °C or >86° F; cold environment: <0 °C or <32° F; high-altitude environment: >3,050 m or >10,000 ft elevation) was used, and three extreme environmental categories were established.
Previous studies have reported that the amount of energy and nutrients required differ depending on the extreme environment (*Tassone & Baker, 2017*). A report by *Tharion et al. (2005)* showed that energy consumption did not tend to be affected by hot environments, but it tended to increase in cold and high-altitude environments. Meanwhile, a prospective cohort study by *Gan et al. (2022)* evaluated changes in energy and physical performance during ranger training (62 days) in a hot and humid environment in 17 male army personnel. The total daily energy expenditure (TDEE) estimated by the double-labeled water method was 4,756 kcal/day, while the average energy intake was 3,882 kcal/day, indicating an energy deficit of 874 kcal/day (18%). In addition, body weight decreased by 4.6 (2.6) kg, or 6.7 (3.8%), over 62 days, with some members losing up to 10.1 kg (14.9%), which also showed a decline in performance evaluation. *Ahmed et al. (2019)* evaluated dietary intake under four conditions: (1) exercise (as standardized infantry activities) in the heat (30 °C), (2) exercise in the cold ($-10$ °C), (3) exercise in temperate thermoneutral (21 °C) air temperatures and (4) a resting (sedentary) trial (21 °C). As in other previous studies (*Johnson et al., 2018*), the results indicated that even with a significant increase in energy expenditure and exposure to severe stress due to temperature, the energy intake from the diet of the military remained the same as at rest, leading to an energy deficit, which is problematic. Therefore, it is very important to consider not only physical activity but also extreme environmental conditions.

Although there are various methods for formulating the energy consumption and basal metabolic rate, the same estimation formula as the DRI for civilians (*Henry, 2005*; *IOM, 1997*; *Schofield, 1985*) was used as a reference. These estimation formulas used reference body size that reflected the physical characteristics of the military members. (Table S3) (*Department of Defense, 2017*; *Forbes-Ewan & DSTO, 2009*; *NATO, 2010*; *Scientific Advisory Committee on Nutrition, 2016b*) Australia, the UK, and the USA used the physical characteristics of their own military members as reference body size, while NATO used the reference body size of American male members (*Baker-Fulco et al., 2001*; *Department of Defense, 2001*). As mentioned above, previous research has identified the lack of energy among military personnel as an important issue (*Barringer et al., 2018*; *Murphy et al., 2018*; *O'Leary, Wardle & Greeves, 2020*; *Vyas & Cialdella-Kam, 2020*). This also suggests that the accumulation of data on the physical characteristics and basal metabolic rate of military personnel, which is common to all four published DRIs, is an issue for the future. Hence, the future challenge for all four published DRIs for military operations were to accumulate data on military personnel's physical characteristics and basal metabolic rates (*Department of Defense, 2017*; *Forbes-Ewan & DSTO, 2009*; *NATO, 2010*; *Scientific Advisory Committee on Nutrition, 2016b*). The UK uses the same equation to estimate energy requirements for both civilian and military populations (*Henry, 2005*); although the aim is to derive an equation specific for military DRI once sufficient data are available (*Scientific Advisory Committee on Nutrition, 2016b*; *SACN, 2020*).

### Protein

The required amount of protein was formulated by reflecting the military's PAL in all four published DRIs for military operations (*Department of Defense, 2017*; *Forbes-Ewan*

& DSTO, 2009; NATO, 2010; Scientific Advisory Committee on Nutrition, 2016b), which differed from the DRI for civilians (Table S4) (Department of Health, 1991; IOM, 1997; NHMRC, 2017). Table S4 shows "Protein in Dietary Reference Intakes (DRI) for civilians and military (/day)". Australia, the USA, and NATO listed the intake weight (g or g/kg) (Department of Defense, 2017; Forbes-Ewan & DSTO, 2009; NATO, 2010), while the UK only listed the percentage (%) of macronutrient energy distribution (Scientific Advisory Committee on Nutrition, 2016b). These are developed with PAL and military environments in mind; the RDA for protein for civilians is 45–65 g/day (Department of Health, 1991; IOM, 1997; NHMRC, 2017), whereas for military values 1.2 to 4.8 times higher than RDA for civilians were established (55–307 g/day) (Department of Defense, 2017; Forbes-Ewan & DSTO, 2009; NATO, 2010; Scientific Advisory Committee on Nutrition, 2016b). These reasons include the need to increase intake according to the PAL category (Chapman et al., 2020; Chapman et al., 2021; Tarnopolsky et al., 2005) and, at the same time, from the perspective of preventing increased urinary calcium excretion and kidney disease due to excessive protein intake (Fouque, Laville & Boissel, 2006). In reports on athletes and sports enthusiasts, it has been argued that protein intake should be a balance between "effectiveness for high performance" and "safety to avoid the risk of kidney damage" (Marinaro, Alexander & de Waal, 2021).

### Macronutrient energy distribution

For the macronutrient energy distribution, the four published DRIs for military operations (Department of Defense, 2017; Forbes-Ewan & DSTO, 2009; NATO, 2010; Scientific Advisory Committee on Nutrition, 2016b) had different energy ratios (%) than those for civilians (Department of Health, 1991; IOM, 1997; NHMRC, 2017; Scientific Advisory Committee on Nutrition, 2011; Scientific Advisory Committee on Nutrition, 2015) (Tables S4, S5, S6, S7 and S8). The USA and NATO recommended a balance with a high proportion of carbohydrates in high-altitude environments (Department of Defense, 2017; NATO, 2010).

Fats differed from DRI for civilians (Department of Health, 1991; IOM, 1997; NHMRC, 2017) in all four published DRIs for military operations (Table S5) (Department of Defense, 2017; Forbes-Ewan & DSTO, 2009; NATO, 2010; Scientific Advisory Committee on Nutrition, 2016b). In Australia and the UK, DRI for civilians were used as a reference, but as the number of PALs increased, the ratio decreased for each category (Forbes-Ewan & DSTO, 2009; Scientific Advisory Committee on Nutrition, 2016b). In the USA, energy consumption was set at less than 30%; therefore, it was within the range of DRI for civilians (Acceptable Macronutrient Distribution Range; AMDR) (Department of Defense, 2017). NATO reflected the same energy ratio as DRI for civilians (NRV) (NATO, 2010).

Carbohydrate intake is important when considering optimal performance at varying levels of physical activity and within extreme environments, such as heat, cold, and high–altitude. Therefore, Australia, the UK, and the USA have set higher standards depending on PAL and environmental extremes (IOM, 1996; Rodriguez, Di Marco & Langley, 2009; Tarnopolsky et al., 2005). Accordingly, the DRI for military operations (Department of Defense, 2017; Forbes-Ewan & DSTO, 2009; Scientific Advisory Committee on Nutrition, 2016b) in these three countries was 1.0–5.2 times higher than the DRI for civilians (Table S7)

(*Department of Health, 1991*; *IOM, 1997*; *NHMRC, 2017*; *Scientific Advisory Committee on Nutrition, 2015*). In NATO, the need to increase carbohydrate standards was considered in personnel with high physical activity (*Tarnopolsky et al., 2005*), but no changes were made to the energy ratio of carbohydrates in DRI for military operations, as an increase in the PAL category would inevitably increase carbohydrate intake without an increase in the energy ratio of carbohydrates (*NATO, 2010*). Therefore, NATO's DRI (NIV) was set at approximately 1–1.7 times the NRV (*NHMRC, 2017*) of the DRI for civilians, lower than the standards for the other three countries (*NATO, 2010*).

The macronutrient energy distribution in Australia and the UK was formulated based on the DRI for civilians (*Department of Health, 1991*; *NHMRC, 2017*; *Scientific Advisory Committee on Nutrition, 2015*), such that as the PAL category increased, the proportion of energy from protein decreased while that from carbohydrates increased, and fat changes were minimal (*Department of Defense, 2017*; *Scientific Advisory Committee on Nutrition, 2016b*). Additionally, to prevent acute mountain sickness (*Askew, 2004*; *IOM, 1996*), Australia and NATO have established a category with a high carbohydrate ratio in high-altitude environments (*Forbes-Ewan & DSTO, 2009*; *NATO, 2010*). Hence, the energy ratio from each macronutrient should consider the PAL for optimal energy balance, with the following specific recommendations. (1) Set the intake protein weight based on the above matters; (2) The upper limit for lipids should be the energy ratio for civilians; (3) carbohydrate intake should be set to increase as PAL increases, and should be high in high-altitude environments (*Department of Defense, 2017*; *Forbes-Ewan & DSTO, 2009*; *NATO, 2010*; *Scientific Advisory Committee on Nutrition, 2016b*).

### Dietary fiber

For dietary fiber, the same DRI was used for civilians (*Department of Health, 1991*; *IOM, 1997*; *NHMRC, 2017*; *Scientific Advisory Committee on Nutrition, 2015*) and was reflected in DRI for military operations in Australia, the UK, and NATO (Table S9) (*Forbes-Ewan & DSTO, 2009*; *NATO, 2010*; *Scientific Advisory Committee on Nutrition, 2016b*). The USA also stated in a survey report (*Department of Defense, 2017*) that the DRI for civilians (*IOM, 1997*) was to be reflected; however, a keen analysis of the survey revealed a different adequate intake (AI) (DRI: male 30 g/day, female 21–25 g/day (*IOM, 1997*); MDRI: male 34 g/day, female 28 g/day (*Department of Defense, 2017*)). Presumably, the total carbohydrate intake has increased; thus, the recommended amount of dietary fiber has also been added.

### Vitamin B$_1$, vitamin B$_2$, niacin, and vitamin B$_6$

For vitamin B$_1$, excluding Australia, the DRI for civilians (*Department of Health, 1991*; *IOM, 1997*; *NHMRC, 2017*) is also reflected in the DRI for military operations (Table S10) (*Department of Defense, 2017*; *NATO, 2010*; *Scientific Advisory Committee on Nutrition, 2016b*). Australia (*Forbes-Ewan & DSTO, 2009*) recommends an increase in vitamin B$_1$ of 0.1 mg per unit of energy (MJ) rather than following the DRI for civilians (*NHMRC, 2017*).

For vitamin B$_2$, the DRI for civilians (*Department of Health, 1991*; *IOM, 1997*) was reflected in the UK and USA (Table S11) (*Department of Defense, 2017*; *Scientific Advisory Committee on Nutrition, 2016b*). Although the DRIs in Australia and NATO were based on the recommended dietary intake (RDI) for civilians (*NHMRC, 2017*), they were developed

for each PAL category and extreme environment for military activities (*Forbes-Ewan & DSTO, 2009*; *NATO, 2010*). Australia recommended to increase vitamin $B_2$ by 0.15 mg per energy (MJ) (*Forbes-Ewan & DSTO, 2009*). In NATO (*NATO, 2010*), 2.5 mg/day was formulated for combat or special forces operations and in extreme environments, which is 1.9 times the NRV of the DRI for civilians (*NHMRC, 2017*).

The DRI for niacin for civilians (*Department of Health, 1991*; *IOM, 1997*; *NHMRC, 2017*) was reflected in the DRI for military operations, except in Australia (Table S12) (*Department of Defense, 2017*; *NATO, 2010*; *Scientific Advisory Committee on Nutrition, 2016b*). Australia recommended to increase niacin by 1.6 mg per energy (MJ) (*Forbes-Ewan & DSTO, 2009*).

Vitamin $B_6$ was formulated in Australia and NATO for each PAL category and extreme environment for military activities, although it was based on the RDI of the DRI for civilians (*NHMRC, 2017*) (Table S13) (*Forbes-Ewan & DSTO, 2009*; *NATO, 2010*). Australia formulated the per PAL category as 0.02 mg per g of protein (*Forbes-Ewan & DSTO, 2009*). In combat or special forces operations and extreme environments in NATO, (*NATO, 2010*) 2.6 mg/day was recommended, double the DRI for civilians (*NHMRC, 2017*).

Vitamin $B_1$, vitamin $B_2$, niacin, and vitamin B6 were added to the recommended amounts in Australia and NATO due to the increased energy and protein intake.

### Sodium, iron, zinc, copper

Due to a lack of scientific evidence, most micronutrients reflect the same DRIs as for civilians; however, sodium has been formulated at more than twice the DRI for civilians in Australia, the USA, and NATO (Table S14) (*Department of Defense, 2017*; *Forbes-Ewan & DSTO, 2009*; *NATO, 2010*). Many previous studies have reported that sodium intake affects the risk of high blood pressure and coronary heart disease (*Xue et al., 2020*). Therefore, in the DRI for civilians, indicators such as the upper level of intake (UL), tolerable upper intake level (UL), and lower reference nutrient intake (LRNI) have been established to set upper limits (*IOM, 1997*; *NHMRC, 2017*). However, the DRI for military operations (*Department of Defense, 2017*; *Forbes-Ewan & DSTO, 2009*; *NATO, 2010*) has considered that engaging in military activities leads to increased sweating and sodium loss in sweat (*IOM, 1996*; *Tomczak et al., 2016*). The NRV of the DRIs for Australia and NATO civilians are AI 460–920 mg/day and UL 2,300 mg/day (*NHMRC, 2017*). The DRI for Australia military operations was increased from 920–23,00 mg/day to 920–3,200 mg/day for increasing PAL categories, with <4,600 mg/day recommended for personnel not accustomed to hot environments (*Forbes-Ewan & DSTO, 2009*). NATO recommended a DRI of 920 mg/day and supplementation of 1,200–4,800 mg, depending on perspiration rate, for personnel engaged in military activities during practice or operations and in extreme environments (*NATO, 2010*). In the USA, the UL in the DRI for civilians (2005) (*IOM, 1997*) was reflected in the MDRI (<2,300 mg/day) (*Department of Defense, 2017*). Thus, the three DRIs for military operations had formulated values that were more than twice as high, but were within the range of DRIs for civilians as the scientific evidence was still lacking and did not consider the physique and physical activity level of the military personnel (*Department of Defense, 2017*; *Forbes-Ewan & DSTO, 2009*; *NATO, 2010*).

Although there are limited studies on sodium requirements for military personnel, there have been reports on athletes (*Baker et al., 2016*; *Barnes et al., 2019*; *Rollo et al., 2021*). *Barnes et al. (2019)* evaluated sweat production and sodium loss in 1,303 athletes from various sports from 2000 to 2017 and reported significant differences among events in whole-body sweating rate. Individual differences were observed; however, American football, endurance sports (runners, cyclists, and triathletes), basketball, and soccer had the greatest sweat loss, respectively, and baseball players having the least. Therefore, as more research reports are accumulated, it is expected that details of the recommended amount of sodium for athletes and military personnel will be established.

For iron, zinc, and copper, the DRI of NATO alone was based on the DRI for civilians (*NHMRC, 2017*), while the DRI for nutrients was established based on military activities and extreme environments (Tables S15–S17) (*NATO, 2010*). NATO's DRI was 1–1.83 times greater than the DRI for civilians for iron, 1–1.43 times greater for zinc, and 1–1.06 times greater for copper. For NATO, the recommended zinc (+7%), iron (+75%), and copper (+6%) intakes were higher for combat than during normal operations (*NATO, 2010*). They reported that sweating associated with increased physical activity would sufficiently increase sweat losses of zinc, iron, copper, and sodium to merit a compensatory increase in daily intake during combat operations.

### Information on DRI for military operations in Asia

We organized information on Singapore, China, and South Korea, where we deemed insufficient information based on the formulation (Table S18). Similar to the four published DRIs for military operations (*Department of Defense, 2017*; *Forbes-Ewan & DSTO, 2009*; *NATO, 2010*; *Scientific Advisory Committee on Nutrition, 2016b*), Singapore and China developed separate categories from those of civilians, reflecting PAL and military activities (*Chan, 2016*; *Guo et al., 2016*; *MND, 2017*). In these three Asian countries (*Chan, 2016*; *Guo et al., 2016*; *MND, 2017*), the added nutrients formulated were as need-added as the DRIs of the four Western countries (*Department of Defense, 2017*; *Forbes-Ewan & DSTO, 2009*; *NATO, 2010*; *Scientific Advisory Committee on Nutrition, 2016b*) formally reviewed, and the scope of the DRIs was similar (Table S18). Based on this, the DRI for military operations (*Department of Defense, 2017*; *Forbes-Ewan & DSTO, 2009*; *NATO, 2010*; *Scientific Advisory Committee on Nutrition, 2016b*) that were officially subject to review can be sufficiently applied to Asians and others after correcting for differences in physique.

## LIMITATIONS AND PERSPECTIVES

### Limitations

First, this review only included English studies, which may have limited the extent of data access. Second, a limited search was applied with one database and a web search engine. Third, the search was performed four years ago (May 31, 2020). Therefore, we encourage readers to also review the new literature on the subject. Fourth, the information on micronutrients in the context of semi-quantitative surveys with questionnaires should only be interpreted with caution, as the risk of incorrect information is high. In addition,

such data do not consider absorption in the body, as the bioavailability of heavy metals, for example, changes greatly whether they are consumed in a fasting state or with food.

**Perspectives**

This study sought to provide consolidated data on military operation DRIs from four nations/organizations. The DRI for military operations that was the subject of this study included survey reports that had been recently revised (*Department of Defense, 2017*; *SACN, 2020*). The reasons for this were that the DRI for civilians had been revised and information such as previous research that was the basis for its formulation was updated. As the DRI for civilians and research reports for the military continue to be updated, it is expected that the DRI for military operations will also be revised. Additionally, a review evaluating the health and performance effects of energy insufficiency in military personnel points to a lack of research and randomized controlled trials that assess long-term effects (*O'Leary, Wardle & Greeves, 2020*). Therefore, as research reports accumulate, nutrients that have not been formulated may be updated.

The DRI for military operations is currently in use in reports evaluating the dietary intake of military personnel (*Ahmed et al., 2023*; *Chapman et al., 2020*; *Garron & Klein, 2023*; *Lutz et al., 2019*) and in studies targeting athletes (*Heaney et al., 2010*) and firefighters (*Johnson & Mayer, 2020*). The DRI for military operations is designed primarily for rations and combat ration packs; however, since it was developed based on high physical activity, it is also expected to be applied for other high-physical activity occupations and athletes.

# CONCLUSIONS

In this study, we investigated DRIs for military operations, compared with those for civilians. Specifically, we reviewed the DRIs for the military operations of four countries and organizations (Australia, the UK, the USA, and NATO). The results demonstrate the necessity for considering DRIs that differ from those of civilians, focusing on energy, protein, macronutrient energy distribution, and sodium. As the DRI recommendations continue to evolve, it is necessary to include dietary fiber, B vitamins, and other micronutrients currently lacking evidence. Furthermore, it is necessary to consider DRIs according to environmental factors such as heat, cold, and high altitude.

The findings summarized in this scoping review will prove useful to meal providers and dieticians for military personnel, athletes, and those in occupations requiring high physical activity and special environments. However, in developing DRIs for military operations, scientific rationale, including a literature review, will continue to be necessary.

### Funding

This study is part of the "Research Service for the Revision of the Standards of Nutritional Intake," funded by Acquisition, Technology & Logistics Agency. Moreover, this work was supported by the Practical Research Project for Lifestyle-related Diseases Including Cardiovascular Diseases and Diabetes Mellitus from the Ministry of Health, Labour and Welfare (Grant Number 22FA1004 to Motohiko Miyachi). The funders had no role in study design, data collection and analysis, decision to publish, or preparation of the manuscript.

### Grant Disclosures

The following grant information was disclosed by the authors:
Acquisition, Technology & Logistics Agency.
Practical Research Project for Lifestyle-related Diseases Including Cardiovascular Diseases and Diabetes Mellitus from the Ministry of Health, Labour and Welfare: 22FA1004.

### Competing Interests

The authors declare there are no competing interests.

### Author Contributions

- Ryoko Mizushima conceived and designed the experiments, performed the experiments, analyzed the data, prepared figures and/or tables, authored or reviewed drafts of the article, and approved the final draft.
- Motohiko Miyachi conceived and designed the experiments, authored or reviewed drafts of the article, management duties for the entire research project, and approved the final draft.
- Eiichi Yoshimura conceived and designed the experiments, authored or reviewed drafts of the article, and approved the final draft.
- Yoichi Hatamoto conceived and designed the experiments, authored or reviewed drafts of the article, and approved the final draft.
- Mai Matsumoto conceived and designed the experiments, performed the experiments, authored or reviewed drafts of the article, and approved the final draft.
- Yuka Hamada performed the experiments, analyzed the data, authored or reviewed drafts of the article, and approved the final draft.
- Mana Hatanaka performed the experiments, analyzed the data, authored or reviewed drafts of the article, and approved the final draft.
- Aya Maeno performed the experiments, analyzed the data, authored or reviewed drafts of the article, and approved the final draft.
- Chifumi Shimomura performed the experiments, analyzed the data, authored or reviewed drafts of the article, and approved the final draft.
- Hidemi Takimoto conceived and designed the experiments, authored or reviewed drafts of the article, management duties for the entire research project, and approved the final draft.

## Data Availability

This is a literature review.

## Supplemental Information

Supplemental information for this article can be found online at http://dx.doi.org/10.7717/peerj.18353#supplemental-information.

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
