# Peer review of "Dietary reference intake for military operations: a scoping review"

_PeerJ, doi:10.7717/peerj.18353_

## Round 0.1 · original submission · Major Revisions

We have received two extensive reviews from two experts in this field, which are very critical. Please be aware that this was a borderline decision. If you decide to revise the manuscript, all comments from both reviewers would really need to be well addressed, and both reviewers would need to agree again in the next review round.

Reviewer 1 ·

Basic reporting

This scoping review deals with a very important and current issue. Due to the changed global geopolitical security situation an increasing risk of armed conflicts and war can be assumed. In addition, the risk of natural and technological disasters is also increasing due to effects of climate change, where the armed forces are mainly involved with subsidiary assistance. For these reasons, the topic "dietary reference intakes for military operation" is more topical than ever! Because of the wide use of the RDIs worldwide, it is important to illustrate their limitations and applications by discussing the method that have been used to assess requirements, especially under difficult conditions such as in military service. The physiological requirement (inclusive physical activity and other important factors) for a nutrient should be the basis for calculating a reference intake.

Unfortunately, research and therefore the literature in this area is very limited. Nevertheless, I am not satisfied with the way the authors have dealt with the subject in this review, despite the high number of authors listed.

- The text, especially the introduction, is very difficult to read because of too many references for a scoping review, some of which are outdated and of lesser importance. Moreover, many redundant references. This manuscript cites >100 references and >50 references are older than 10 years. Fifty well-chosen and up-to-date references are sufficient for such a review. The authors should reconsider the references and reduce them drastically.

- I don't understand why different places in the text are marked with red text. I think it is most likely in the context of previous revisions in another journal.

- Lines 86 and 87: This sentence can be deleted, as neither the results nor the discussion deals with this topic: "In addition, eating an appropriate diet over a long period is expected to reduce the risk of obesity and cardiovascular disease".

- Line 130: "……..for military operations in each country and region and……". In this sentence the word "each" must be replaced with "several" or "some", because this review only analyses a considerable number of states.

Experimental design

- Lines 131-133: The aim of the present review "These findings summarized in this review will be useful for meal providers to military personnel, athletes, and those in occupations requiring high physical activity and special environments, as well as dietitians who support their activities" is clearly described, but all these hypotheses are not sufficiently and in depth addressed, neither in the results section nor in the discussion. In this regard, the authors have to revised this aspect along the whole manuscript.

- Lines 143-153: I have two important concerns regarding the methodology section: A) The search in PubMed was carried out in 2020. For such a review, I expect the search to be no older than one year. Why this latency of four years between search and submission for publication? B) the authors simply used the database PubMed and Google, which is more web search engine than a library database, to search for relevant articles. The Optimal searches in good quality reviews should search at least Embase, Web of Science, Scopus and Google Scholar as a minimum requirement to guarantee adequate and efficient coverage. In addition, only eight articles were found in PubMed, of which only one study was included, which is itself a scoping review.

Validity of the findings

- Sections results and discussion: Very much characterized by the search results and not actually by interesting data on the main topic of the review. Thanks to the Collins et al. study (Collins et al. 2020), some important data are reproduced. For me, the sections results and discussion are very confusing without a clear structure. Data from various publications are simply quoted without the authors carrying out their own reflection and in-depth analysis and placing this data in the context of the main topic. The authors' own analysis and discussion of the collected data is insufficient.

- Lines 192-193: The authors write the following sentence: " A keen study of this scoping review by 192 Collins et al.[32] resulted in the discovery of survey reports of dietary intake standards in the 193 military in the USA[61-63] and Australia[70] being cited as references". It is very unusual to make two kinds of citations in the same manuscript. The numbers in square brackets cannot be looked up in the present review, so please cite other sources or delete them.

- Lines 275-277: The authors state the following: "Energy and nutrient indicators are important, and there are various indicators (Aggett et al. 1997; King et al. 2007). Hence, there are various ways to formulate and understand them (Table S1). Therefore, we organized the data based on the type of indicator (Table 3)". These indicators cannot be found in either the text or the table. Can the authors specify these indicators and also assess them according to sensitivity and specificity?

- The chapters about "Energy requirements and PAL categories", " Proteins", "Dietary fiber", "Vitamin B1, Vitamin B2, Niacin and Vitamin B6" and "Sodium, Iron, Zinc, Copper" are not well structured and the discussion remains superficial without the authors' own reflection. For example, in the case of energy and protein intake, it would make sense to state these as kcal (mj) or g per kilo of body weight per day, as a total figure of one quantity per day says nothing about the individual weight-adapted requirement and is therefore also difficult to compare between the studies. The information on micronutrients in the context of semi-quantitative surveys with questionnaires should only be used and interpreted with caution, as the risk of incorrect information is high. In addition, such data say nothing about the actual absorption in the body, as the bioavailability of heavy metals, for example, changes greatly whether they are taken in fasting state or with food. I would therefore recommend concentrating on energy and macronutrients in this paper and delving deeper into the topic when interpreting and discussing the data according to the principle "less is more".

- Limitations: Following statements are missing: A) Only studies in English were included which may have limited the extent of data access. B) This review has a limited search with only one database and with a web search engine. C) The search presented is already for years old.

Additional comments

- The conclusions lack clear statements and recommendations by the authors as stated in the detailed hypothesis.

- Acknowledgment: This text section is mixed with the declaration of conflict of interest. Please take both apart and state them separately.

Overall, this work has several weak points as mentioned above. The authors have dealt with the topic too superficially and have not analysed and discussed it enough in depth. They also lack important statements and recommendations of their own, as well as an outlook for future research.

·

Basic reporting

Your manuscript is interesting and can be value-added to the military community. A scoping review is a tedious process, so thank you for your efforts to illustrate the current evidence and areas for future research.

The introduction and methods sections were predominantly easy to read; however, the results / discussion, limitations / perspectives, and conclusions were difficult to read due to sentence structure and flow. An editorial service could assist. I've included specific recommendations for your consideration to improve clarity, concision, and coherence of content, that should help reduce ambiguity for the reader. Many times citations were unnecessarily included in the middle of sentences creating broken thoughts and thoughts lost in translation while scanning several lines of references before finishing the sentence.

Abstract:
Lines 39-40: For concision and clarity, I recommend revising to “The PAL range for the military was set as high as 1.5-3.2, as opposed to the civilian upper PAL set at 1.2-2.2.”
Line 40: Grammatically, I recommend inserting “of” between “outside the”.

Introduction:
Line 100: when stating “this review”, the reader may think “this” refers to your current scoping review. To clarify, I recommend changing to “their” or “Collins et al.”.
Lines 114-115: For concision, I recommend revising the sentence to “A recent study reported using the 2017 version of the MRDI (….) to assess dietary intake of military personnel.”
Line 127: For concision, I recommend grammatical edits (removing commas) and revise to “… who perform high physical activity levels similar to military personnel.”

Experimental design

A few consideration regarding study design:
1) Please address how you obtained the civilian DRI source material. I assume you followed a similar search strategy as that for the military. This should be addressed with the methodology since you are directly comparing and you are not the author of these source documents.
2) Consider that the "Army" is only one of 4 military services within the USA. Your search strategy should include Air Force, Navy, and Marines. That being send AR 40-25 was ultimately published in conjunction with Navy (OPNAVINST 10110.1), Air Force (AFI 44–141), and Marines (MCO 10110.49), but there may be other reports or documents of relevance that you missed by limited your search terms to “army”. You should mention these source documents and perhaps run a quick search to ensure you haven't missed other key documents.
3) Be cautious and clear with the use of your terminology. You repeatedly state "four DRI" but it is not clear if you mean the four nations/organizations that published DRIs, or are speaking of four specific nutrient DRIs (especially considering your abstract and summary focus on 4 DRI components).
4) There is an incomplete explanation of how you took all of your raw data depicted within the Supplementary Tables, and consolidated to yield your four final tables. After rereading a few times, and reviewing your supplementary tables, the content made more sense. I recommend that you add a short paragraph to describe the transition from your raw data supplementary tables to your finished consolidated tables at line 263. For example, “Physical activity levels (PAL) category definitions and estimated energy requirement calculations vary among nations/organizations by age, sex, height, weight, and activity level (Supplementary Tables S2-S3). Specific energy and nutrient DRIs for civilian and military populations by nation/organization are depicted in Supplementary Tables S4-S18. These supplementary tables were used to develop the consolidated framework needed to illustrate differences in DRIs between civilian and military populations provided in Table 3 and PAL categories in Table 4.”

Specific Recommendations:
Line 138: I recommend a colon after “was” before starting your list, and adding a semicolon or a period after each listed sentence.
Line 147: I recommend doing a quick search using the terms “air force”, “navy”, and “marines” to ensure these services are represented as they are distinct military services of the US Department of Defense. The AR 40-25 is jointly published with Navy (OPNAVINST 10110.1), Air Force (AFI 44–141), and Marines (MCO 10110.49), but there may be other reports or documents of relevance that you missed by limited your search terms to “army”.
Line 161: The term ‘dietary reference intake’ is missing the end apostrophe.
Line 178: I recommend removing the parenthesis symbol to read, “In Step 2, we…”
Line 181: You mention DRI for the civilian population but don’t mention how you acquired them. I assume this was actually part of your search strategy to identify the most up-to-date relevant civilian-based DRIs to compare with the military operation DRIs that you found. Please address.

Validity of the findings

With revisions to sentence structure for clarity, concision, and coherence, the validity of your findings will be more transparent to the reader. You've performed a lot of work to pull this article together and need to tidy up the translation regarding rationale and implications. I've included some recommendations (see additional comments section).

Overall: I struggled with several areas of your results for several reasons:
1) As a reader I often had to scan several lines to find the end of your sentence. For example, line 240 you pause the sentence to add cited referenced before finishing your thoughts on line 247 with “were organized (Table 1).” Please move up those last few words to complete the sentence prior to the listing your references. Consider this throughout your results/discussion.
2) There is a lack of explanation regarding the transition between supplementary raw data to the finished consolidated tables. After rereading a few times, and reviewing your supplementary tables, the content made more sense. I recommend that you add a short paragraph to describe the transition from your raw data supplementary tables to your finished consolidated tables at line 263. For example, “Physical activity levels (PAL) category definitions and estimated energy requirement calculations vary among nations/organizations by age, sex, height, weight, and activity level (Supplementary Tables S2-S3). Specific energy and nutrient DRIs for civilian and military populations by nation/organization are depicted in Supplementary Tables S4-S18. These supplementary tables were used to develop the consolidated framework needed to illustrate differences in DRIs between civilian and military populations provided in Table 3 and PAL categories in Table 4.”
3) Consider deleting supplementary Table S1 as it does not add much value and is just another way to organize the information, of which Table 2 is better. I found Supplementary Tables S2-S18 very helpful to understand your results/discussion and thus throughout section 3 you should consistently include the Supplementary Table number that is associated with the consolidated data discussed.

Line 234: I’m assuming that you are referring to the nations/organizations who published the DRIs. To reduce confusion throughout the paper, I recommend when discussing 4 DRI that you specify “four nation/organizations DRI” or “four published DRIs” to make clear that you are not discussing the DRIs for four specific nutrients (especially considering that your abstract summarizes 4 nutrient-based results. Please review all areas where you state “four DRI” to ensure the correct context (i.e., line 238, 262, 263, and beyond).
Line 261: What does “Basic matters for” mean? I am wondering if by ‘formulating’ you mean organizing information. Please consider word choice for clarity. I was very confused with your topic sentence (lines 263-267). As mentioned above inserting a short paragraph to explain how supplementary data was used to develop your Tables 3-4 would be helpful to the reader.
Line 271: I recommend replacing “corps” with “military” because not all military services are organized by corps. For example, the Air Force organizes their airman in flights, squadrons, and wings.
Line 273: My interpretation of Table 2 is not that the countries differ in intakes between military and civilian, but that the nomenclature of indicators used varies. Please clarify.
Line 275: You state indicators are important. How so? I’m not sure this sentence adds value and could be deleted or Lines 275-277 should be revised. Perhaps you mean to say: “Research depicts that energy and nutrient indicators are formulated and displayed in a variety of ways. Table 3 present the difference in indicator nomenclature to better compare each across the four organizations.” But referring back to my earlier comment about describing how the supplementary tables are precursors to consolidated tables, may rectify the problem here. Either way, please clarify for the reader.
Lines 280-281: I am not sure what this sentence means or the implications. Please clarify what the difference is and why it matters. Also, as mentioned early, the list of citations added in the middle of a sentence is very burdensome to the reader. In this case, moving the only word remaining in your sentence (“index”) to immediately following “NRV” allows the reader to complete the thought.
Line 284: As written this statement is ambiguous and a bit confusing with the 2 concepts blended together. The first half of the topic sentence on line 285 leads the reader to believe you will talk about Table 3 - do you mean “The estimated energy requirements for military operations among 3 countries and NATO are congruent with their civilian DRI for total energy and the three macronutrients (protein, fat, and carbohydrates) as depicted in Table 3”(?).
Then the 2nd half of the sentences (line 287) jumps right into PAL levels in relation to energy DRI difference. I recommend splitting Energy requirements and PAL categories into 2 separate sections. If my interpretation is incorrect, please clarify.
Line 299: For clarity, I recommend inserting “of the four nation/organizations” after “..,operations” and “ranges from similar…” to clarify that you are contrasting the four published DRIs.
Line 314: your examples (Soldier, Sailors, Airmen, and Marines) are not considered occupations, but instead they are descriptors of what each services calls their members. The military has many occupations that mirror that of the civilian sector (healthcare, administrative, management, supply chain, education, police, firefighters, etc.) but also has occupations with higher physical that may not translate well to a civilian occupation (i.e., infantry, cavalry). To prevent confusion, I recommend revising your sentence to something similar to “energy expenditures measured in 424 military members across all four military services, in a variety of occupations, climates, and environments (field vs. garrison) ranged from …” and I recommend deleting the end of the sentence: “with a wide range that the range of MJ/day was very wide” as this statement is circular and the reader can see the huge variability with the values presented for males/females in lines 316-317.
Lines 321-322: the end of the sentence “… it is assumed that…” is speculative and inconsequential to the content. It is more accurate to state that these special operators are trained to function for short periods of time with reduced sleep and intense physical exertion.
Line 328-330: please define the variables that correspond to “cut-off values” earlier in the sentence. For example, “…environmental cut-off values…were used”.
Line 334: Grammatically, “differs” should be “differ”.
Line 335: You state “William et al.” but cite Tharion et al. Please double check and revise.
Line 337: You state “Linda et al.” but then cite Gan et al. Please double check and revise.
Line 344: You state four conditions but five are listed. Please double check and clarify.
Line 354-5: This is a good place to add “(Supplementary Table S3)” at the end of one of these two sentences.
Line 368: The sentence is awkward to read. Consider revising to something similar to “The UK uses the same equation to estimate energy requirements for both civilian and military populations; although the aim is to derive an equation specific for military DRI once sufficient data are available.” If I missed your intent, please clarify.
Line 373: As you continue to discuss each nutrient, I recommend that you provide the specific Supplementary Table # somewhere in each section. This prompts the reader if needed.
Line 385: There is no need to capitalize “Value” and a word or two is missing in this phrase “whereas for military values 1.2 to 4.8 times higher than RDA for…”
Line 395: Since protein also provides energy, this is a little confusing. Perhaps rename this heading to “Macronutrient Energy Distribution” because you address all three macronutrients that provide energy.
Line 410: Recommend replacing the colon with a semicolon.
Line 414: It is not accurate to say “Carbohydrates were considered in PAL”. Instead, do you mean, carbohydrate intake is important when considering optimal performance at varying levels of physical activity and within environmental extremes?
Line 415: To ensure clarity for the reader, ensure that you specify that you are talking about carbohydrate (not just DRI). For example, “Therefore, the carbohydrate DRI for military operations was set at 1.0 to 5.2 times higher than that of the civilian population.” Again, as noted earlier, in many cases by moving most citations to the end of the sentence you will dramatically improve reading flow and content synthesis for the reader.
Line 428: As mentioned on line 395, I recommend simplifying “energy-producing nutrient balance” to “macronutrient energy distribution” as this is the accepted terminology in the nutrition field.
Line 430: To clarify, I recommend revising to “such that as PAL category increased, the proportion of energy from protein decreased while carbohydrate increased, and fat changes were minimal.”
Lines 436-439: The sentence is awkward to read. To clarify: “Based upon these results, the energy ratio from each macronutrient should consider the PAL for optimal energy balance, with specific recommendations as follows: …”
Sections 3.14-3.16. I noticed that for some nutrients you provided thoughts about the relevance or role of that nutrient to the military when considering why the DRI values differed from civilians (energy, protein, carbs, fat, sodium), but not for all (i.e., no discussion for fiber or the B vitamins), except where you state there is limited evidence.
Line 516: I recommend deleting (WBSR) because it is only used once. Too many acronyms can be tedious for the reader. Additionally, this sentence is very long with many commas. Consider breaking into 2 sentences.
Line 525: You mentioned that iron, zinc, and copper DRIs were considered for military operations. What was the result? Were their difference or not? The reader can refer to the supplementary tables for specific detail, but will look to the authors to provide a statement.
Lines 528-542 (Section 3.2): This paragraph was unclear. Partially due to citations in the middle
of the sentence disrupting the flow and partially due to the structure of thoughts within the sentences. There is circular logic that does not make sense on line 536: “needed added nutrients formulated were as need-added as DRI”). As the reader, I am unable to paraphrase the intent of this information. Please reread, consider your intent and revise this section.

Limitations and Perspectives:
Line 545: I recommend flipping your sentence clauses for clarity: “The aim of this study was to provide consolidated data on military operation DRIs from four nations/organizations.”
You discuss your perspective but need to address the limitations of your methodology. I mentioned earlier, one limitation incomplete search terms to ensure all US military services are represented. Consider other potential limitations based upon your assumptions when extracting and consolidating the extensive information.
Lines 555-56: These sentences are better suited earlier in the paragraph. Consider organizing thoughts: 1) Purpose of study; 2) DRIs already in use when designing rations; 3) DRIs considered in research / reports to assess the quality of dietary intake in a variety of military operations; 4) differences in how DRIs are formulated, many of which mirror that of the civilian population without accounting for PAL of military operations; lack of evidence for all nutrients; 5) lack of evidence/research for all nutrients to promotion military performance; and future research can fill this gap.

Conclusions:
Line 566: I recommend removing “its outline with” and replacing with “against the”.
Line 568: Again I recommend revising “3) Energy-providing nutrient balance” to “Macronutrient energy distribution” (as well as in your abstract).
Line 569-570: Please clarify this circular logic “… when it becomes necessary formulating these stands, it is necessary to review the standard…” I think your intent is: “As the DRI recommendations continue to evolve, it is necessary to include dietary fiber, the B vitamins, and other micronutrients currently lacking evidence.”

Additional comments

Table 1:
Title: The table is not an overview of the DRIs but instead the timeline of development and updates to DRIs. Recommend changing the title to something more accurate, i.e., “Overview of the development and update timeline for civilian and military DRI.”
Footnote: “DRI for civilians based on each DRI for military operations” is inaccurate. This reads as if the civilian DRIs are derived from those set from military operations. Do you mean, “The civilian DRIs were used as the precursor for DRI development for military operations”? Please clarify.

Table 2:
Title: The overview is not only for USA and Canada (by the way, this is the first time that I’ve seen Canada included), but for Australian, UK and NATO. Consider a title that captures all nations / organizations included.
Column headings: I recommend revising your main column heading “Dietary Reference Intakes; DRI” to improve the tables ability to stand alone. For instance, “Dietary Reference Intake (DRI) Indicator Acronym and Description”
Description: Because the acronym in in the preceding column, you can delete “The EAR is”, The RDA is”, etch for each row. You can simply start with “An average daily…”, etc.
Not applicable row: Because you have plenty of room in the “Other” box, I recommend inserting those three acronyms with a symbol to direct reader to the footnote for a definition. “SDT+, LRNI++, DRV+++”
Abbreviations: As written, it is difficulty for the reader to searching for a specific acronym in your list due to its organization. I recommend: 1) list them all in alphabetical order; and 2) put the acronym first followed by a comma, with a semicolon signifying the next acronym. For example, “AI, Adequate Intake; AMDR, Acceptable Macronutrient Distribution Range; BMI, Body Mass Index; …., etc.”

titles: As tables are stand alone, it is recommended that you spell out any acronyms in the title (i.e. PAL should be spelled out).

Table 3:
Title: For accuracy, I recommend replacing “different from” with “comparison to”.
Footnote: You state “the nutrients organized in this review were assumed to be common to…” What does that mean?

Table 4:
Title: Please spell out PAL in the title in order to improve its ability to stand alone (same for any of your supplementary tables).
UK: Are you able to discern what is the difference between Military Training Courses A & B. I assume it is the intensity, such as initial vs. advanced training. If available, I recommend adding to the footnotes.
Footnotes: I recommend that symbolled be arranged in the order they appear: UK *, USA +, NATO ++

---

## Round 0.2 · accepted · Accept

Thank you for addressing all of the reviewers comments. I am pleased to recommend your amended manuscript for publication.

Reviewer 1 ·

Basic reporting

This manuscript deals with a very important and topical issue, which deserves to be published as the literature on the subject is scarce.
I am satisfied with the changes and corrections made by the authors. The authors have made an effort and the main points of criticism have been addressed and implemented.
Reducing the number of references to the essentials will also make the manuscript easier to read and understand.

Experimental design

Not everything could be implemented in the methodology, but since the literature on this topic is sparse, I agree with the current result of the revision.

Validity of the findings

PeerJ's readership will certainly find the content of great interest. The conclusions are well stated.

Additional comments

Overall, I think the manuscript is now good in terms of structure and content.

·

Basic reporting

Revisions to the manuscript improved clarity and concision.

Experimental design

Revisions acceptable. No further comments.

Validity of the findings

Revisions sufficient. No additional comments.

Additional comments

Thank you for taking the time to revise your manuscript. It was a pleasure to read and a value-added contribution to the military and scientific community.